# STRUCTURED RAG FOR ANSWERING AGGREGATIVE QUESTIONS

## ABSTRACT

Retrieval-Augmented Generation (RAG) has become the dominant approach for answering questions over large corpora. However, current datasets and methods are highly focused on cases where only a small part of the corpus (usually a few paragraphs) is relevant per query, and fail to capture the rich world of *aggregative queries*. These require gathering information from a large set of documents and reasoning over them. To address this gap, we propose S-RAG, an approach specifically designed for such queries. At ingestion time, S-RAG constructs a structured representation of the corpus; at inference time, it translates natural-language queries into formal queries over said representation. To validate our approach and promote further research in this area, we introduce two new datasets of aggregative queries: HOTELS and WORLD CUP. Experiments with S-RAG on the newly introduced datasets, as well as on a public benchmark, demonstrate that it substantially outperforms both common RAG systems and long-context LLMs.

## 1 INTRODUCTION

Retrieval-Augmented Generation (RAG) has emerged as a leading approach for the task of Open Book Question Answering (OBQA), attracting significant attention both in the research community and in real-world applications (Lewis et al., 2020; Guu et al., 2020; Yoran et al., 2023; Ram et al., 2023; Izacard et al., 2023; Gao et al., 2023; Siriwardhana et al., 2023; Fan et al., 2024). Most prior work has focused on *simple* queries, where the answer to a given question is explicitly mentioned within a short text segment in the corpus, and on *multi-hop* queries, which can be decomposed into smaller steps, each requiring only a few pieces of evidence.

While RAG systems made substantial progress for the aforementioned query types, the task of answering *aggregative queries* still lags behind. Such queries require retrieving a large set of evidence units from many documents and performing reasoning over the retrieved information. Consider the real-world scenario of a financial analyst tasked with answering a question such as, 'What is the average ARR for South American companies with more than 1,000 employees?'. While such a query could be easily answered given a structured database, it becomes significantly harder when the corpus is private and unstructured. In this setting, RAG systems cannot rely on the LLM's parametric knowledge; instead, they must digest the unstructured corpus and reason over it to generate an answer, introducing several key challenges: Information about the ARR of different companies is likely to be distributed across many documents, and even if the full set of relevant evidence is retrieved, the LLM must still perform an aggregative operation across them. Moreover, aggregative queries often involve complex filtering constraints (e.g., 'before 2020', 'greater than 200 kg'), which vector-based retrieval systems often struggle to handle effectively (Malaviya et al., 2023).

Current RAG systems handle aggregative questions by supplying the LLM with a textual context that is supposed to contain the information required to formulate an answer. This context is constructed either by retrieving relevant text units using vector-based representations, or by providing the entire corpus as input, leveraging the extended context windows of LLMs. Both strategies, however, face substantial limitations in practice. Vector-based retrieval often struggles to capture domain-specific terminology, depends on document chunking and therefore limits long-range contextualization, and requires predefining the number of chunks to retrieve as a hyperparameter (Weller et al., 2025). Conversely, full-context approaches are restricted by the LLM's context size and its limited long-range reasoning capabilities (Xu et al., 2023).

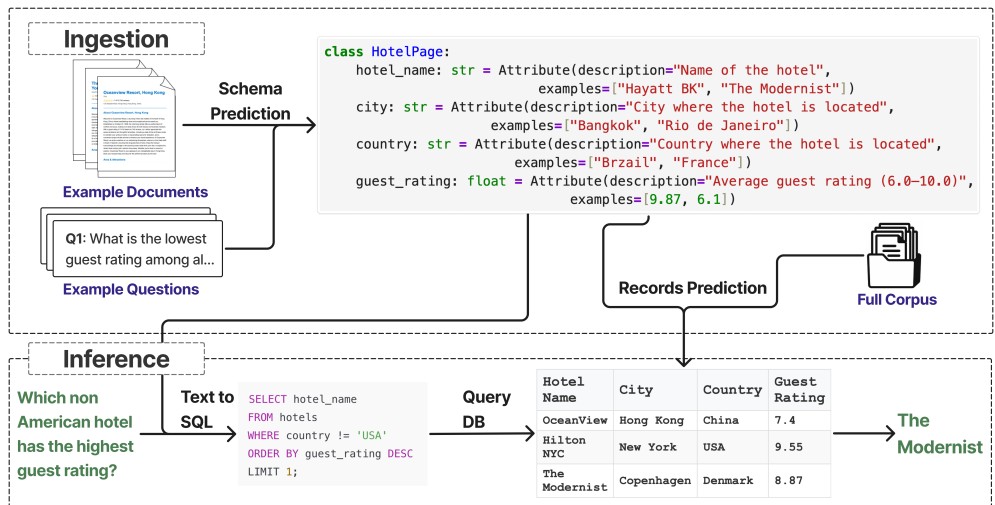

Figure 1: S-RAG overview. Ingestion phase (upper): given a small set of questions and documents, the system predicts a schema. Then it predicts a record for each document in the corpus, populating a structured DB. Inference phase (lower): A user query is translated into an SQL query that is run on the database to return an answer.

In this work, we introduce Structured Retrieval-Augmented Generation (S-RAG), a system designed to address the limitations of existing techniques in answering aggregative queries over a private corpus. Our approach relies on the assumption that each document in the corpus represents an instance of a common entity, and thus documents share recurring content attributes. During the ingestion phase, S-RAG exploits those commonalities. Given a small set of documents and representative questions, a schema that captures these attributes is induced. For example, in a corpus where each document corresponds to a hotel, the predicted schema might include attributes such as hotel name, city, and guest rating. Given the prediction, each document is mapped into an instance of the schema, and all resulting records are stored in a database. At inference time, the user query is translated into a formal language query (e.g., SQL), which is run over the ingested database. Figure 1 illustrates the ingestion phase (in the upper part) and inference phase (in the lower part).

To facilitate future research in this area, we introduce two new datasets of aggregative question answering: (1) HOTELS: a fully synthetic dataset composed of generated booking-like hotel pages, alongside aggregative queries (e.g., 'What is the availability status of the hotel page with the highest number of reviews?'); and (2) WORLD CUP: a partially synthetic dataset, with Wikipedia pages of FIFA world cup tournaments as the corpus, alongside generated aggregative questions. Both datasets contain exclusively aggregative questions that require complex reasoning across dozens of text units.[1]

We evaluate the proposed approach on the two newly introduced datasets, as well as on FinanceBench (Islam et al., 2023), a public benchmark designed to resemble queries posed by financial analysts. Experimental results demonstrate the superiority of our approach compared to vector-based retrieval, full-corpus methods, and real world deployed services.

To conclude, our main contributions are as follows:

1. We highlight the importance of aggregative queries over a private corpus for real-world scenarios and demonstrate the limitations of existing benchmarks and methods in addressing this challenge.

2. We introduce two new datasets, HOTELS and WORLD CUP, specifically designed to support future research in this direction.

3. We propose a novel approach, S-RAG, for handling aggregative queries, and show that it significantly outperforms existing methods.

---

[1]The datasets are publicly available at: https://www.anonymous.com

## 2 AGGREGATIVE QUESTIONS OVER UNSTRUCTURED CORPUS

Retrieval-augmented generation (RAG) has become the prevailing paradigm for addressing the Open-Book Question Answering (OBQA) task in recent research (Gao et al., 2023; Asai et al., 2024; Wolfson et al., 2025), and it is now widely adopted in industrial applications as well. Substantial progress has been made in answering simple queries, for which the answer is explicitly provided within a single document. In addition, considerable effort has focused on improving performance for multi-hop questions, which require retrieval of only a few evidence units per hop (Yang et al., 2018; Trivedi et al., 2022; Tang & Yang, 2024). Despite this progress, aggregative questions, where answering a question requires retrieval and reasoning over a large collection of evidence spread across a large set of documents, remain relatively unexplored.

Yet aggregative questions are highly relevant in practical settings, especially for organizations working with large, often unstructured, private collections of documents. For instance, an HR specialist might query a collection of CVs with a question such as 'What is the average number of years of education for candidates outside the US?'. Although the documents in such a corpus are written independently and lack a rigid structure, we can assume that all documents share some information attributes, like the candidate's name, years of education, previous experience, and others.

Standard RAG systems address the OBQA task by providing an LLM with a context composed of retrieved evidence units relevant to the query (Lewis et al., 2020; Ram et al., 2023). The retrieval part is typically performed using dense or sparse text embeddings. Such an approach would face several challenges when dealing with aggregative queries:

1. **Completeness**: Failing to retrieve a single required piece of evidence might lead to an incorrect or incomplete answer. For example, consider the question 'Who is the youngest candidate?' – all of the CVs in the corpus must be retrieved to answer correctly.

2. **Bounded context size**: Since the LLM context has a fixed token budget, typical RAG systems define a hyper-parameter $K$ for the number of chunks to retrieve. Any question that requires integrating information from more than $K$ sources cannot be fully addressed. Furthermore, the resulting context might be longer than the LLM's context window.

3. **Long-range contextualization**: Analyst queries often target documents with complex structures containing deeply nested sections and subsections (e.g., financial reports). Consequently, methods that rely on naive chunking are likely to fail to capture the full semantic meaning of such text units (Antropic, 2024).

4. **Embedders limitation**: As shown by (Weller et al., 2025), there are inherent representational limitations to dense embedding models. Furthermore, sparse and dense embedders are likely to struggle to capture the full semantic meaning of filters (Malaviya et al., 2023), especially when handling named entities to which they were not exposed at training time.

## 3 S-RAG: STRUCTURED RETRIEVAL AUGMENTED GENERATION

This section describes S-RAG, our proposed approach for answering aggregative questions over a domain specific corpus. Similarly to vector-based retrieval, we suggest a pipeline consisting of an offline Ingestion phase (§3.2) and an online Inference phase (§3.3). See Figure 1 for an illustration.

### 3.1 PRELIMINARIES

Consider a corpus $D = \{d_1, d_2, \ldots, d_n\}$ of $n$ documents, where each document $d_i$ corresponds to an instance of an entity, described by a schema $\mathcal{S} = \{a_1, a_2, \ldots, a_m\}$, where each $a_j$ denotes a primitive attribute with a predefined type. For example, in a corpus of CVs, the entity type is a CV, and the underlying schema may include attributes such as an integer attribute 'years of education' and a string attribute 'email'. For each document $d_i$, we define a mapping to its *record* $r$:

$$r(d_i) = \{(a_j, v_{ij}) \mid a_j \in \mathcal{S}\}, \tag{1}$$

where $v_{ij}$ is the value of attribute $a_j$ expressed in document $d_i$. Importantly, the value $v_{i,j}$ may be empty in a document $d_i$. An aggregative question typically involves examining $a_j$ and the corresponding set $\{v_{1,j}, v_{2,j}, \ldots, v_{n,j}\}$, optionally combining a reasoning step. This formulation can be naturally extended to multiple attributes. Figure 2 illustrates our settings.

Figure 2: Illustration of a naive CVs corpus, schema and a single record. An example of an aggregate query on such a corpus could be: 'Which candidates has more than two years of experience?'

## 3.2 INGESTION

The ingestion phase of S-RAG aims to derive a structured representation for each document in the corpus, capturing the key information most likely to be queried. This process consists of two steps:

### 3.2.1 SCHEMA PREDICTION

In this step, S-RAG predicts a schema $\mathcal{S} = \{a_1, a_2 \ldots a_m\}$ that specifies the *entity* represented by each document in the corpus. The schema is designed to capture recurring attributes across documents, i.e. attributes that are likely to be queried at inference time. We implement this stage using an iterative algorithm in which an LLM is instructed to create and refine a JSON schema given a small set of documents and questions. The LLM is prompted to predict a set of attributes, and to provide for each attribute not only its name but also its type, description, and several example values. The full prompts used for schema generation are provided in Appendix B.[2] We do zero-shot prompting with 12 documents and 10 questions, quantities tailored for real-world use cases, where a customer is typically expected to provide only a small set of example documents and queries.

### 3.2.2 RECORD PREDICTION

Given a document $d_i$ and a schema $\mathcal{S}$, we prompt an LLM to predict the corresponding record $r_i$, which contains a value for each attribute $a_j \in \mathcal{S}$. The LLM is provided with the list of attribute names, types, and descriptions, and generates the output set $\{v_{i,1}, v_{i,2}, \ldots, v_{i,m}\}$. Each predicted value $v_{i,j}$ is then validated by post-processing code to ensure it matches the expected type of $a_j$.

Since the meaning of a value $v_{i,j}$ can be expressed in multiple ways (e.g., the number one million may appear as 1,000,000, 1M, or simply 1), attribute descriptions and examples are crucial for guiding the LLM in lexicalizing $v_{i,j}$ (e.g., capitalization, units of measure). Because the same descriptions and examples are shared across the prediction of different records, this process enables cross-document standardization.

After applying this prediction process to all documents in the corpus $D$, we store the resulting set of records $\{r_1, r_2, \ldots, r_n\}$ in an SQL table. Finally, we perform post-prediction processing to compute attribute-level *statistics* based on their types (more details are provided in Appendix D). These statistics are used at inference time, as detailed next.

## 3.3 INFERENCE

At inference time, given a free-text question $q$, an LLM is instructed to translate it into a formal query over the aforementioned SQL table. To enhance the quality of the generated query and avoid ambiguity, the LLM receives as input the query $q$, the schema $\mathcal{S}$ and statistics for every column in the DB. These statistics guide the LLM in mapping the semantic meaning of $q$ to the appropriate lexical filters or values in the formal query. The resulting query is executed against the SQL table, and the output is stringified and supplied to the LLM as context.

---

[2]For simplicity at inference time, we exclude list and nested attributes, since these would require reasoning over multiple tables.

**Hybrid Inference Mode** When the predicted schema fails to capture certain attributes, particularly rare ones, the answer to a free-text query cannot be derived directly from the SQL table. In such cases, we view our system as an effective mechanism for narrowing a large corpus to a smaller set of documents from which the answer can be inferred. To support this use case, we experimented with HYBRID-S-RAG, which operates in two inference steps: (i) translating $q$ into a formal query whose execution returns a set of documents (rather than a direct answer), and (ii) applying classical RAG on the retrieved documents.

## 4 AGGREGATIVE QUESTION ANSWERING DATASETS

While numerous OBQA datasets have been proposed in the literature, most of them consist of simple or multi-hop questions (Abujabal et al., 2018; Malaviya et al., 2023; Tang & Yang, 2024; Cohen et al., 2025). To support research in this area, we introduce two new aggregative queries OBQA datasets: HOTELS and WORLD CUP. The former is fully synthetic, containing synthetic documents and questions, while the latter contains synthetic questions over natural documents.

### 4.1 AGGREGATIVE DATASETS CREATION METHOD

To create a dataset of aggregative questions, we start by constructing a schema $\mathcal{S}$ that describes an entity (e.g., hotel). $\mathcal{S}$ consists of $m$ attributes (e.g. city, manager name, etc.), each defined by a name, data type, and textual description. We then generate $n$ records of $\mathcal{S}$ by employing an LLM or code-based randomization. Each generated record corresponds to a distinct entity (e.g., Hilton Paris, Marriott Prague). We then apply LLMs in two steps: (1) given a structured record $r_i$, verbalize its attributes into a natural language html document $d_i$ (see Appendix C); and (2) given a random subset of records, formulate an aggregative query over them and verbalize it in natural language.

### 4.2 HOTELS AND WORLD CUP DATASETS

**Hotels.** This dataset is constructed around hotel description pages, where each entity $e$ corresponds to a single hotel. Each page contains basic properties such as the hotel name, rating, and number of stars, as well as information about available facilities (e.g., swimming pool, airport shuttle). An example document is provided in Appendix C. Using our fully automatic dataset generation pipeline, we produced both the documents and the associated question-answer pairs. Our document generation process ensures that some of these properties are embedded naturally within regular sentences, unlike other unstructured benchmarks, which often present properties in a table or within a dedicated section of the document (Arora et al., 2023). The resulting dataset consists of 350 documents and 193 questions. We consider this dataset to be more challenging, as public LLMs have not been exposed to either the document contents or the questions.

**World Cup.** This dataset targets questions commonly posed within the popular domain of international soccer. The corpus consists of 22 Wikipedia pages, each corresponding to one of the FIFA World Cup tournaments held between 1930 and 2022. To increase the difficulty of the corpus, we removed the main summary table from each document, as it contains structured information about many key attributes. Based on this corpus, we manually curated 22 structured records and used the automatic method described in §4.1 to generate 83 aggregative questions. Although LLMs are likely to possess prior knowledge of this corpus, evaluating RAG systems on these aggregative questions provides an interesting and challenging benchmark.

Table 1 summarizes the statistics of the introduced datasets. It also compares them to FINANCEBENCH (Islam et al., 2023), a public benchmark designed to resemble queries posed by financial analysts. In contrast to our new datasets, questions in FinanceBench typically require up to a single document to answer correctly (usually a single page).

## 5 EXPERIMENTAL SETTINGS

### 5.1 BASELINES

We implement VECTORRAG, a classic embedder based approach. It performs chunking and dense embedding at ingestion time, followed by chunk retrieval using a dense embedder at inference time

Table 1: Statistics and characteristics of datasets used in our experiments.

| Dataset | # Documents | Avg. Tokens / Doc | # Queries | Aggregative | LLM leak? |
|---|---|---|---|---|---|
| Hotels | 350 | 596 | 193 | High | ✗ |
| World Cup | 22 | 18881 | 88 | High | ✓ |
| FinanceBench | 360 | 109592 | 150 | Low | ✓ |

(see Appendix A). We note that VECTORRAG is on-par with the best performing method reported by Wang et al. (2025) on FINANCEBENCH, and therefore we consider it as a well performing system.

In addition, we provide results of FULLCORPUS pipeline, in which each document is truncated to a maximum length of 20,000 tokens. The context is then constructed by concatenating as many of these document prefixes as can fit within the LLM's context window.

We also report the performance of a real-world deployed system, OPENAI-RESPONSES by OpenAI (OpenAI, 2025). This agentic framework supports tool use, including the FileSearch API. Although it is a broader LLM-based system with capabilities extending beyond RAG, we include it in our evaluation for completeness. Unlike the baselines we implemented, OPENAI-RESPONSES is a closed system that directly outputs the answer, limiting our control on its internal implementation.

## 5.2 S-RAG VARIANTS

S-RAG is evaluated in three settings: (i) **S-RAG-GoldSchema**: skip the Schema Prediction phase, and provide an oracle schema to S-RAG. This schema contains all the relevant attributes to answer all of the queries in all aggregative benchmarks, (ii) **S-RAG-InferredSchema**: predict schema based on a small set of documents and queries which are later discarded from the dataset, and, (iii) **HYBRID-S-RAG**: as explained in §3.3, we use S-RAG to narrow down the corpus and perform VECTORRAG over the resulting sub-corpus.

## 5.3 ANSWER GENERATOR

Every RAG system includes an answer generation step, in which an LLM generates an answer given the retrieved context and the input question. For S-RAG, we employ GPT-4o for this step. In contrast, for the baselines VECTORRAG and FULLCORPUS, we use GPT-o3 with stronger reasoning capabilities. This ensures fairness, since in our setting the reasoning steps are handled in SQL, while in the baselines the LLM must perform them. In addition, to minimize the influence of the model's prior knowledge, we explicitly instructed the LLM in all experiments to generate answers solely on the basis of the provided context, disregarding any external knowledge.

## 5.4 EVALUATION DATASETS

We evaluate S-RAG on the two newly introduced datasets, HOTELS and WORLD CUP, as well as on the publicly available evaluation set of FINANCEBENCH. Since the FINANCEBENCH test set includes both aggregative and non-aggregative queries, we report results on the full test set as well as on the subset of 50 queries identified by the original authors as aggregative[3].

In order to estimate the familiarity of existing LLMs with our evaluation sets, we build a context-less question answering pipeline, where a strong reasoning model was asked to answer the question without any provided context. Table 2 shows the performance of GPT-o3 in this setting. As expected, GPT-o3 fails on HOTELS as it includes newly generated documents, but surprisingly achieves an AnswerComparison score of 0.71 on WORLD CUP. We consider the results on HOTELS as evidence that only a robust pipeline can succeed on this dataset, while the strong performance on WORLD CUP likely reflects the familiarity of modern LLMs with Wikipedia content.

---

[3]Referred to as the "metrics-generated queries"

Table 2: Zero-shot performance of o3 without any provided context.

| Dataset | Answer Recall | Answer Comparison |
|---|---|---|
| FinanceBench | 0.443 | 0.505 |
| Hotels | 0.047 | 0.049 |
| WorldCup | 0.798 | 0.712 |

## 5.5 METRICS

Following prior work on evaluating question answering systems, we adopt the *LLM-as-a-judge* paradigm (Zheng et al., 2023). Specifically, to compare the expected answer with the system generated answer, we define two evaluation metrics: (1) **Answer Comparison**, where the LLM is instructed to provide a binary judgment on whether the generated answer is correct given the query and the expected answer (metric meta-analysis and prompt are provided in Appendix E); and (2) **Answer Recall**, where an LLM-based system decomposes the expected answer into individual claims and computes the percentage of those claims that are covered in the generated answer. For both metrics, we employ GPT-4o as the underlying judging model.

## 6 RESULTS

Table 3 summarizes the results of S-RAG and the baselines when evaluated on the aggregative questions evaluation sets. Across all datasets, S-RAG consistently outperforms the baselines[4], although those systems employ a strong reasoning model when possible.

**FULLCORPUS:** All datasets exceed GPT-o3's context window, and therefore it can't process the full corpus directly (which is a major difference compared with Wolfson et al. (2025)). As expected, this baseline fails to achieve strong results on any dataset. HOTELS is relatively smaller, leading to reasonable performance, but a real-world use cases involve much larger corpora.

**VECTORRAG & OAI-RESPONSES:** Results for both VECTORRAG and OAI-RESPONSES are reasonable ($\sim$10-20% behind S-RAG-GoldSchema) when parametric knowledge is available (FINANCEBENCH, WORLD CUP), however, it falls short on HOTELS ($\sim$50-60% behind S-RAG-GoldSchema). As discussed in §2, vector-based retrieval suffers from inherent limitations when considering aggregative questions. This is most prominent with HOTELS, where the generating model is unable to compensate suboptimal retrieval with parametric knowledge. This also holds for OAI-RESPONSES, even though it is able to execute multiple retrieval calls, which exemplifies the *completeness* issue (the backbone model cannot tell when to stop the retrieval).

**S-RAG-InferredSchema:** For simpler documents, like the generated HOTELS, or Wikipedia pages of WORLDCUP tournaments, our system is solid, which leads to overall strong performance. There is a degradation in performance compared with GoldSchema. This stems from failures in the schema prediction phase, specifically: (i) missing attributes; (ii) incomplete descriptions which lead to standardization issues in the DB. This problem intensifies with complex documents such as in FINANCEBENCH, leading to poor performance. For example, we saw that the CapitalExpenditure attribute was described as "The capital expenditure of the company". Thus, in the record prediction phase (§3.2.2) two values were recorded as 1, but one of them stands for 1M and the other for 1B which makes it unusable at inference time. However, given that manually building the gold schema via prompting required only a few hours, we regard this as a practical and feasible approach for real-world applications.

**S-RAG-GoldSchema:** Best results are achieved across datasets when providing the gold schema. The imperfect scores can be attributed to imperfect text-to-sql conversion, standardization issues in the ingestion phase, and wrong records prediction.

Finally, Table 4 shows the performance of HYBRID-S-RAG with gold schema on the full FINANCEBENCH, including aggregative and non-aggregative queries. The superior results of HYBRID-S-RAG demonstrate that S-RAG can perform well also on general purpose datasets.

---

[4]When baseline methods are employed with GPT-4o as the underlying LLM, their performance is even lower than the results reported in the table.

Table 3: Results of different systems on the aggregative evaluation sets.

| Dataset | System | Ingestion Type | Answer Recall | Answer Comparison |
|---|---|---|---|---|
| Hotels | VectorRAG | — | 0.352 | 0.331 |
| | FullCorpus | — | 0.478 | 0.473 |
| | OAI-Responses | — | 0.253 | 0.184 |
| | S-RAG | InferredSchema | 0.500 | 0.518 |
| | S-RAG | GoldSchema | **0.845** | **0.899** |
| WorldCup | VectorRAG | — | 0.735 | 0.676 |
| | FullCorpus | — | 0.516 | 0.441 |
| | OAI-Responses | — | 0.715 | 0.566 |
| | S-RAG | InferredSchema | 0.766 | 0.769 |
| | S-RAG | GoldSchema | **0.909** | **0.856** |
| FB-Agg | VectorRAG | — | 0.650 | 0.598 |
| | FullCorpus | — | 0.100 | 0.040 |
| | OAI-Responses | — | 0.670 | 0.593 |
| | S-RAG | InferredSchema | 0.230 | 0.234 |
| | S-RAG | GoldSchema | **0.750** | **0.725** |

Table 4: Performance on the full FinanceBench evaluation set.

| System | Answer Recall | Answer Comparison |
|---|---|---|
| VECTORRAG | 0.598 | 0.677 |
| OAI-RESPONSES | 0.529 | 0.553 |
| HYBRID-S-RAG | **0.667** | **0.702** |

Table 5: An example from WORLD CUP dataset with answers generated by different systems.

**Question:** *What is the average number of total goals scored across all World Cups in this dataset?*

**Gold Answer:** *The average number total goals scored across all World Cups is approx. 123.64.*

| System | Generated Answer | Explanation |
|---|---|---|
| VECTORRAG | *Adding the total-goal for every World Cup the context lists: 1950 – 88, . . . ,2022 – 172; Total goals = 1,739 tournaments = 13 Average = 1,739 ÷ 13 ≈ 133.8* | The context includes information on only 13 tournaments; the 1954 World Cup, for instance, is missing. |
| FULLCORPUS | *Across the eight World Cup tournaments cited:•1934: 70 goals . . . • Total goals = 70 + 126 + . . . + 171 = 954 Average = 954 ÷ 8 ≈ 119.25* | The context includes only eight tournament documents, as context length of the LLM is limited. |
| S-RAG | *Average = 123.64* | S-RAG produces a short and correct answer based on the information from the twenty two tournaments. |

**Qualitative Examples.**    Table 5 presents the answers generated by different systems for the natural aggregative query, 'What is the average number of total goals scored across all World Cups in this dataset?', from the WORLD CUP dataset. Both VECTORRAG and FULLCORPUS produce the wrong answer: despite the reasonable reasoning chain, the incomplete context results in an incorrect answer. In contrast, S-RAG delivers a concise and accurate answer, demonstrating its performance on aggregative queries that require reasoning over a large set of evidence across multiple documents.

# 7 RELATED WORK

## 7.1 RAG SYSTEMS

Modern RAG systems typically address the Open-Book Question-Answering task by retrieving the text units from the corpus that are most relevant for answering the query according to some relevance score (Lewis et al., 2020; Ram et al., 2023). At the ingestion phase, a standard system splits each document independently into a set of chunks and computes a vector representation for each chunk. These representations are obtained either through sparse embeddings (Robertson et al., 2009), which represent text as high-dimensional and interpretable vectors based on explicit lexical features, or dense embeddings (Muennighoff et al., 2022; Wang et al., 2022), which encode text into low-dimensional continuous vectors that capture semantic similarity, enabling effective retrieval even when queries and documents share little lexical overlap. The retrieval phase is typically carried out by scoring the relevance of each chunk to the query, using their vector representations, and optionally applying post-retrieval re-ranking on the top scoring chunks, utilizing a model that jointly encodes the chunk and the query.

In addition to domain-agnostic approaches, corpus-specific training has also been explored, for example by Wang et al. (2025), though such methods suffer from limited scalability. Among structure-based methods, Edge et al. (2024) propose constructing a knowledge graph at ingestion time to capture information essential for answering queries. However, their approach is primarily designed for global sense-making questions and is not built to handle aggregative queries (as it does not enforce a recurring structure in the graph which is the cornerstone of such queries). Another noteworthy contribution is by Arora et al. (2023), who propose building structured representation of an unstructured corpus. Nevertheless, their system was not evaluated in the context of RAG performance.

## 7.2 OPEN-BOOK QA DATASETS

Most existing OBQA datasets include simple questions for which the answers are explicitly contained within an individual text segment of the corpus, or require reasoning over no more than a handful of such evidence pieces (Nguyen et al., 2016; Abujabal et al., 2018; Yang et al., 2018; Trivedi et al., 2022; Malaviya et al., 2023; Tang & Yang, 2024; Cohen et al., 2025). This tendency arises as annotating questions and answers is considerably easier when focusing on small number of text units. Others construct questions that require the integration of a larger number of evidence units (Wolfson et al., 2025; Amouyal et al., 2023); however, these datasets do not focus on large-scale retrieval, and are based on Wikipedia, a source which LLMs are well exposed to during pretraining. This underscores the need for new datasets that require multi-document retrieval over unseen corpora, while also involving diverse reasoning skills such as numerical aggregation.

# 8 CONCLUSIONS

In this work, we highlight the importance of aggregative questions, which require retrieving and reasoning over information distributed across a large set of documents. To foster further research on this problem, we introduce two new aggregative questions datasets: WORLD CUP and HOTELS. To address the challenges such datasets pose, we propose S-RAG, a system that transforms unstructured corpora into a structured representation at ingestion time and translates questions into formal queries at inference time. This design addresses the limitations of classic RAG systems when answering aggregative queries, enabling effective reasoning over dispersed evidence.

Our work has a few limitations: First, our approach is limited to corpora that can be represented by a single schema, whereas in the real world a corpus may contain documents derived from multiple schemas. In addition, the schemas underlying the datasets we experiment with include only simple attributes, and we encourage future research on corpora that incorporate more complex structures.

In our experiments, S-RAG achieves strong results on the newly introduced datasets and on the public FINANCEBENCH benchmark, even compared to top-performing RAG methods and advanced reasoning models. We further show that the schema prediction step plays a critical role in end-to-end performance, highlighting an important direction for future research.

To conclude, our work puts emphasis on aggregative queries, a crucial, realistic blindspot of current RAG systems, and argues that unstructured, classical methods alone are ill-suited to address them. By introducing new datasets tailored to evaluate such queries, and designing a structured solution, we hope to pave the way to next generation RAG systems.

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

## A  VECTORRAG IMPLEMENTATION DETAILS

The VECTORRAG implementation is as follows:

At ingestion time, each document is split into non-overlapping chunks of 500 tokens, and the `Qwen2-7B-instruct` embedder[5] is applied to obtain dense representations for each chunk. We store each chunk along with its embedded representation in an `Elasticsearch` index.

At inference time, given a query $q$, we use the same embedder to encode the query and retrieve the top 40 chunks with the highest similarity scores. The retrieved chunks are concatenated into a single context, with each chunk separated by a special delimiter token. We do not incorporate a sparse retriever (e.g., BM25) or re-ranking modules, as preliminary experiments showed that they did not yield performance improvements across datasets.

## B  SCHEMA PREDICTION IMPLEMENTATION DETAILS

We run the iterative algorithm for four iterations, employing `GPT-4o` as the underlying LLM. The prompts we used in the schema generation phase are:

---

**Schema generation prompt - first iteration**

```
Task: Extract a single JSON schema from the provided
documents. I'll provide you with a set of documents.
Your task is to analyze these documents and identify recurring
concepts. Then, build a single JSON schema that exhaustively
captures *all* these concepts across all documents.

Focus specifically on identifying patterns that
appear consistently across multiple documents.

Present your response as a complete JSON schema with the
following structure:

```json
{
  "title": "YourSchemaName",
  "type": "object",
  "properties": {
    "fieldName": {
      "type": "string",
      "description": "Detailed description of the field,
      at least two sentences.",
      "examples": ["example1", "example2"]
    }
  },
  "required": ["fieldName"]
}
When building the schema:
- Avoid object-type fields with additional nested properties
when possible.
- Avoid list. Instead use boolean attribute for each of the
potential value.
- Make sure to capture all recurring concepts
- Relevant concepts may include locations, dates, numbers,
strings, etc.
- Relevant concepts should not be lengthy strings (e.g. a
"description" field is not a good choice), you should rather
decompose into separate fields if possible.
```

---

[5]https://huggingface.co/Alibaba-NLP/gte-Qwen2-7B-instruct

648
649
650
651
652
653
654
655
656
657
658
659
660
661
662
663
664
665
666
667
668
669
670
671
672
673
674
675
676
677
678
679
680
681
682
683
684
685
686
687
688
689
690
691
692
693
694
695
696
697
698
699
700
701

**Schema generation prompt - second iteration and on**

```
Task: Refine an existing JSON schema based on set of questions
and documents analysis

I'll provide you with an existing JSON schema, set of questions,
and a set of documents. The JSON schemas of different documents
will be converted into an SQL table, that will be used as knowledge
source to answer questions that are similar to the  provided questions.
Your task is to analyze what attributes from the documents can
provide answers to questions similar to the provided questions,
and refine the existing schema.
Make sure that the attribute value can be extracted (and not
inferred) from each of the documents.

Provide the final refined JSON schema implementation:
```json
{
  "title": "RefinedSchemaName",
  "type": "object",
  "properties": {
    "propertyName": {
      "type": "string",
      "description": "Detailed description of the property,
      at least two sentences.",
      "examples": ["example1", "example2"]
    }
  },
  "required": ["propertyName"]
}

In addition for each attribute and document provide the value
of the attribute in the document.

When evaluating the existing schema:
- Make sure that every property can be extracted from each
of  the documents
- Modify properties where the name, type, or definition could
be improved
- Add new properties for concepts that can help answer the
questions. E.g.: if a question is about "the most common
location", you should add a property for "location" if it
doesn't exist. Make sure that the property value can be
extracted from each of the documents.
- Add new properties for recurring concepts not captured in the
existing schema
- Add new properties for trivial concepts that are missing in
the existing schema. E.g: If the schema represents a house for
sale, it must include the seller's name.
- Use appropriate JSON Schema types (string, number, integer,
boolean, array, etc.)
- Provide descriptions and examples for each property
- Avoid nested object properties
- Fields should not be lengthy strings (e.g. a "description"
field is not a good choice), you should rather decompose into
separate fields if possible.
- Avoid assigning values to the attributes in the schema. You
should only provide the schema itself, without any values.
For each property decision, provide a clear rationale based on
related question or patterns observed in the documents. Your
goal is to create a refined schema that better captures the
recurring patterns that can be used to answer the questions
while minimizing unnecessary changes to the existing structure.
```

## C    EXAMPLE HOTELS DOCUMENT

Example document from the HOTELS dataset:

### The Elegant Chateau, Sydney
Resort

⭐⭐ · 9.21/10 (262 reviews)

123 Luxury Lane, Sydney, NSW 2000, Sydney, Australia

___

**About The Elegant Chateau, Sydney**

Welcome to The Elegant Chateau, a charming 2-star resort nestled in the heart of Sydney, Australia, where comfort meets affordability. Since opening its doors on February 1, 2016, this delightful retreat has garnered rave reviews, boasting an impressive rating of 9.21/10 from 262 satisfied guests. The Elegant Chateau offers a perfect blend of relaxation and leisure, featuring a refreshing swimming pool, a rejuvenating spa, and a soothing sauna to unwind after a day of exploring the vibrant city. Our welcoming bar invites you to indulge in a variety of drinks, while our attentive front desk staff, fluent in English, is always on hand to assist you with your needs. With convenient parking facilities and a prime location that allows easy access to Sydney's iconic attractions, The Elegant Chateau is the ideal choice for travelers seeking a memorable stay. Book your getaway today and experience the perfect balance of comfort and convenience at The Elegant Chateau!

**Area & Attractions**

Nestled just 1 kilometer from the vibrant heart of Sydney's city center, our hotel offers the perfect blend of urban convenience and local charm. Guests can easily explore iconic landmarks such as the Sydney Opera House and the Sydney Harbour Bridge, both within a short stroll or a quick ferry ride from Circular Quay. For those looking to indulge in culinary delights, the nearby Rocks district is a treasure trove of restaurants and cafes, offering everything from fresh seafood to international cuisine. Additionally, the lush expanses of Hyde Park provide a serene escape, perfect for leisurely walks or a picnic under the sun. Conveniently located just 2 kilometers from Sydney Airport, our hotel ensures seamless accessibility for travelers. Whether you're in town for business or leisure, you'll find a plethora of activities nearby, including shopping at the bustling Pitt Street Mall and exploring the art galleries in Surry Hills. With public transport options readily available, getting around the city is a breeze, allowing you to fully immerse yourself in Sydney's vibrant culture and stunning scenery. Experience the best of Sydney right at your doorstep!

**Facilities & Amenities**

| 🏊 Swimming Pool | 🍺 Bar | 💆 Spa | 🔥 Sauna | 🧺 Laundry Service |
| 👔 Dry Cleaning | 🚗 Parking | | | |

**Pricing**

😞 Sold Out

**$407.34** per night

Sold Out     Check Availability

Figure 3: A randomly selected document from the HOTELS dataset

## D    ATTRIBUTE STATISTICS

After applying record prediction to all documents in the corpus, we compute attribute-level *statistics*. For numeric attributes, we calculate the mean, maximum, and minimum values; for string and boolean attributes, we include the set of unique values predicted by the LLM. For all attributes, regardless of type, we also include the number of non-zero and non-null values.

## E    ANSWER COMPARISON JLM

To evaluate the correctness of Answer Comparison JLM, we sampled 112 ⟨question, gold answer, predicted answer⟩ triplets generated by different systems across multiple datasets. A human annotator then provided a binary judgment for each example, indicating whether the predicted answer conveyed the same information as the gold answer. In 90% of the cases, the human annotator agreed with the JLM's assessment. We did not repeat this experiment for Answer Recall JLM, as its scores closely mirror those of Answer Comparison. We provide the following meta analysis examples, each consisting of a question, the gold answer, the system name and its predicted answer, and the corresponding JLM score:

**Answer Comparison Meta Analysis Example 1**

**Question:**
What is the total number of hotels offering laundry service?

**Gold Answer:**
173

**System:**
S-Rag InferredSchema

**Generated Answer:**
The total number of hotels offering laundry service is 269.

**JLM Score:**
0

**Answer Comparison Meta Analysis Example 2**

**Question:**
Answer the following question as if you are an equity research analyst and have lost internet connection so you do not have access to financial metric providers. According to the details clearly outlined within the P&L statement and the statement of cash flows, what is the FY2015 depreciation and amortization (D&A from cash flow statement) % margin for AMD?

**Gold Answer:**
4.2%

**System:**
VectorRag

**Generated Answer:**
Based on AMD's 2015 financials in the excerpts provided: Depreciation & Amortization expense reported in the cash-flow statement (FY-2015) = $167 million.FY-2015 Net revenue (per P&L) = $3,991 million. D&A % margin = $167 ÷ $3,991 ≈ 0.042 (4.2%).Therefore, AMD's depreciation and amortization margin for fiscal-year 2015 is approximately 4.2 percent.

**JLM Score:**
1

**Answer Comparison Meta Analysis Example 3**

**Question:**
In what year did Turkey clinch third place, and which team was the runner-up that year?

**Gold Answer:**
Turkey clinched third place in the year 2002, and the runner-up that year was Germany.

**System:**
OAI-Responses

**Generated Answer:**
Turkey clinched third place in the 2002 FIFA World Cup. That year, Germany was the runner-up, losing to Brazil in the final.

**JLM Score:**
1

---

**Answer Comparison Prompt**

```
<instructions>
You are given a query, a gold answer, and a judged answer.
Decide if the judged answer is a correct answer for the query, based
on the gold answer.
Do not use any external or prior knowledge. Only use the gold answer.
Answer Yes if the judged answer is a correct answer
for the query, and No otherwise.
<query>
{query}
</query>
<gold_answer>
{gold_answer}
</gold_answer>
<judged_answer>
{judged_answer}
</judged_answer>
</instructions>
```

## F  LLM USE

In addition to the uses of LLMs described throughout the paper—for dataset creation, ingestion, and inference—we also employed ChatGPT to help identify mistakes (such as grammar and typos) and to improve the phrasing of paragraphs we wrote.

