# OpenReview forum: "Structured RAG for Answering Aggregative Questions"
_ICLR.cc/2026/Conference — Submitted to ICLR 2026_

### Official Review · Reviewer_b2z7 · 2025-10-29

**Soundness:** 3
**Presentation:** 3
**Contribution:** 3
**Rating:** 6
**Confidence:** 4

**Summary:**

The paper extends the RAG design pattern for document retrieval-augmented question-answering by LLMs to address aggregate queries against information in a corpus of documents, each document of which is a description of a given type of entity. It accomplishes this by mining the corpus to generate a schema for a relational database that can represent the salient features of a given entity as a record, populating a database with the generated schema by extracting a record from a given document, and then translating a natural language aggregate query into a SQL query that can be executed against the generated database.

**Strengths:**

This is an innovative approach to solving a real issue; combining text-to-SQL capabilities with relational data extraction from an entity-centric corpus makes eminent sense. The author(s) contribute two new datasets specifically designed for evaluating aggregative query performance, filling an important gap in existing evaluation benchmarks that typically focus on queries answerable from answers extracted from the top-most relevant documents.

The approach demonstrates strong performance on synthetic data, suggesting the method's potential when its underlying assumptions are met. Additionally, the HYBRID-S-RAG variant shows promise by combining structured extraction for filtering with traditional RAG for final answer generation, achieving competitive performance on the full FinanceBench dataset and suggesting a practical deployment path for mixed query types.

**Weaknesses:**

As the author(s) acknowledge, the technique is currently limited to corpora of entity documents for a single entity type. The approach shows fragility in schema inference, with performance degrading dramatically between gold (i.e., human-curated) and inferred schemas.

The WORLD CUP dataset contains only twenty-two documents while HOTELS uses three hundred fifty synthetic documents, scales far below typical enterprise corpora.

The paper provides no timing comparisons or computational cost analysis for the ingestion phase, leaving open questions about the cost of schema generation and record extraction over large corpora. The aggregative queries tested are relatively simple, focusing on counts and averages rather than complex analytical queries involving multiple joins or nested aggregations that would better demonstrate the approach's capabilities.

**Questions:**

How does the system handle value extraction inconsistencies and normalization? Given that LLMs may extract the same value in different formats such as "1M" versus "1,000,000" versus "one million", what validation, normalization, or post-processing strategies could improve consistency? How robust is the SQL query generation to these variations, and what happens when numeric values are extracted with different units or scales?

What are the practical design constraints for corpus creation? Let's assume for the moment that there is a use case of building corpora that consist of documents each of which describes a given entity of a given type (which this reviewer believes there is). For practitioners wanting to apply this technique, what guidelines could be given for creating such a corpus? This includes considerations around minimum and maximum document length, required attribute coverage across documents, handling of optional versus required fields, and dealing with evolving schemas over time as business needs change.

How does the system handle incremental updates to the corpus? When new documents are added, must the entire schema be regenerated, or can it evolve incrementally? What happens to existing records if the schema changes to accommodate new attributes found in newer documents? How does the system maintain consistency between documents processed at different times?

---

> ### Author Response · Authors · 2025-11-21
>
> We thank Reviewer b2z7 for the time and effort invested in evaluating our paper and for raising valuable questions and concerns. We also appreciate the reviewer’s positive assessment of our proposed method, the recognition of the importance of the newly introduced datasets, and the acknowledgement of the strong performance achieved by the various S-RAG variants on both synthetic and natural data. We now address the raised weaknesses in detail:
>
>
> ## W1: Single Schema Assumption
>
> As this weakness was raised by several reviewers, we kindly refer you to our detailed response here: https://openreview.net/forum?id=jH6K80njJA&noteId=RauDEVf7C4
>
> ## W1: Method Fragility
> We acknowledge that S-RAG-InferredSchema exhibits reduced performance on complex documents, as illustrated by FinanceBench, whose documents are lengthy (over ten pages PDF) and contain tables. This limitation is considerably less pronounced in datasets with shorter and less complex documents. Our analysis indicates that on FinanceBench_AGG, the recall of the Schema Prediction step (measured against the gold attributes) is reasonable - 65% or higher - and that much of the remaining difficulty stems from the standardization issue discussed in the paper.
>
>
> ## W2: Small Scale of Introduced Aatasets
> We assume that the low performance of existing methods on these benchmarks already highlights their limitations in handling aggregative questions. As for the WorldCup dataset - here we were limited to the number of tournaments that took place so far, but still show that although LLMs were trained on this public knowledge, existing QA methods are far from solving this dataset.
> We further expect that enlarging the synthetic datasets would amplify the existing challenges—for example, the FullCorpus method would receive an even smaller proportion of relevant documents as the corpus grows, likely leading to further performance degradation.
>
>
> ## W3: Ingestion Costs and Latency
> We did not report ingestion costs or latency for S-RAG and the competing methods because, in practical deployments, ingestion is typically performed offline and only once per document, and therefore does not influence the user experience. Nonetheless, we expect the ingestion cost and latency of S-RAG to be higher than those of VectorRAG, as S-RAG relies on LLMs to generate a predicted record for each document in the corpus. In contrast, inference latency is an important factor for the end user. At inference time, S-RAG is expected to be more efficient, as it does not rely on a reasoning model as other methods, and is likely to operate on a shorter context, whereas other methods expand the context size irrespective of the specific question.
>
>
> ## Q1: Value Standartization
> We address the value-standardization challenge in section 3.2 and believe that both S-RAG and competing methods are susceptible to such issues, given that documents are authored independently. However, the Schema Prediction step offers an opportunity to enforce consistent value standardization across documents. For example, the inferred schema for FinanceBench_Agg includes the float attribute costOfGoodsSold with the description: “The cost of goods sold (COGS) as reported in the income statement, typically in USD millions.” With this explicit specification of units and the float type, the subsequent record-prediction step for each document is expected to yield standardized values.
> If numeric values are extracted inconsistently, the current implementation of S-RAG cannot correct these discrepancies at inference time. However, we can envision a post–record-prediction procedure in which extreme or outlier values for each attribute are validated against both the document content and the description of the attribute.
>
>
> ## Q2: Corpus Design Choices
> As long as the single-entity constraint holds, S-RAG does not impose any additional requirements on the documents (aside from the need for each document to fit in the LLM’s context window). The datasets we used partially demonstrate this flexibility:
>  (a) the corpus may contain a very large number of documents, since each document is processed independently; (b) optional attributes are permitted, and attributes may be missing from some documents (as in HOTELS); (c) document lengths may vary substantially (e.g., relatively short documents in HOTELS versus very long documents in FinanceBench); (d) the terminology may vary across documents.
>
> ## Q3: Corpus Updates
> When a new document is added to the corpus, only a single record-prediction step is required. This design choice is also reflected in the Schema Prediction step, which operates on a limited subset of documents. S-RAG does not assume schema completeness (i.e., some attributes may be missing), so when a document introduces additional attributes, rerunning the Schema Prediction phase is not necessary. In such cases, Hybrid-S-Rag may help compensate for the missing information.

---

### Official Review · Reviewer_G5VK · 2025-11-01

**Soundness:** 3
**Presentation:** 3
**Contribution:** 3
**Rating:** 4
**Confidence:** 3

**Summary:**

This paper introduces S-RAG, a framework for answering aggregative questions, i.e., queries that require collecting and reasoning over information from many documents. Unlike standard RAG systems that rely on vector retrieval of short passages, S-RAG transforms an unstructured corpus into a structured database at ingestion time. It does so by inducing a schema from sample documents and questions, then predicting structured records for each document via LLM prompting. At inference, user queries are translated into SQL queries over this database, optionally combined with standard RAG in a hybrid mode. To evaluate the proposed framework, authors introduce two aggregative QA datasets. Experiments compare S-RAG to baselines using the two datasets. Results show that S-RAG significantly outperforms these baselines, especially when a gold schema is available.

**Strengths:**

S1) The paper identifies aggregative queries as a distinct and practically important class of QA problems inadequately handled by existing RAG and multi-hop QA systems. Framing this as a structured reasoning challenge is conceptually fresh and well-motivated.

S2) The S-RAG architecture is clearly described, with distinct ingestion and inference phases. The schema-induction process and record standardization pipeline are intuitively presented and easy to follow.

S3) The two new datasets fill a gap by explicitly testing aggregative reasoning over unstructured corpora. Their release could catalyze future work in this space.

S4) Experiments are comprehensive. The quantitative and qualitative examples convincingly demonstrate the failure modes of standard RAG and the strengths of S-RAG in maintaining completeness of evidence.

**Weaknesses:**

O1) While the paper introduces a creative conceptual shift, the technical implementation largely relies on prompting existing LLMs for schema induction, record extraction, and SQL generation. There is minimal methodological innovation beyond careful prompt design.

O2) The entire pipeline hinges on the reliability of LLM-generated schemas and records. These are prone to variability, omission, and inconsistency (as the authors themselves note). Without quantitative analysis of schema accuracy or inter-run variance, it is unclear how stable or reproducible the system is.

O3) The comparison uses different models across systems (e.g., GPT-4o for S-RAG, GPT-o3 for baselines) justified by differing reasoning loads. Although well-intentioned, this complicates claims of superiority—model differences, not the retrieval framework, may partly explain performance gaps. Similarly, since the HOTELS dataset is synthetic and generated with LLMs, data leakage or stylistic biases may favor schema-driven methods.

O4) The use of “LLM as a judge” metrics (Answer Comparison and Answer Recall) introduces subjectivity. The paper does not report agreement statistics or robustness analyses of the judge prompts. There is also no human evaluation to validate correctness, precision, or reasoning quality.

O5) The proposed method assumes that all documents share a single latent schema—a strong constraint that may not hold for most enterprise or web-scale corpora. The authors acknowledge this limitation but do not explore multi-schema or hierarchical settings, which would be essential for real-world applicability.

**Questions:**

Please address Weaknesses O2-O5.

---

> ### Author Response · Authors · 2025-11-17
>
> We thank reviewer G5VK for their time and for their thoughtful, constructive review. We also appreciate your acknowledgment of our strengths—recognizing that we identify an important failure mode of a significant problem, propose an innovative solution, and introduce datasets of substantial importance. We would like to thoroughly respond to the weaknesses raised, and we hope that our answers would satisfy your concerns. We also want to emphasize that we will modify the submission accordingly.
>
> ## W1: Reliance on LLMs
>
> As you said in S1, “the framing of the problem is conceptually fresh”. We believe that relying on prompt design is not a weakness, but a strength, especially given the strong results achieved. Relying on LLMs for various components is a common and well-established practice in recent NLP work. Our proposed conceptual framework, S-RAG, is model-agnostic and can naturally be extended by training dedicated models for each step in the pipeline (e.g., schema prediction, record predictions, or E2E via RLVR methods). For this initial implementation, we employ LLM prompting as the underlying mechanism, allowing us to demonstrate the feasibility and effectiveness of the proposed method. We do believe that future generators could be trained, and we would emphasize this in the discussion section.
>
> ## W2: Schema Prediction
>
> We acknowledge that using LLMs introduces noise, and therefore adds variance to the results. To reduce variance, the Schema Prediction step is implemented as an iterative process (four iterations) where each iteration refines the schema based on another sample of (documents, queries). Such iterations help to reduce variance and improve the consistency of this stage. In addition, both the inferred and manual schemas include detailed guidance on how to lexicalize the attribute value (e.g., “companyName should be in uppercase, with all punctuation removed except periods and commas”), which is expected to further reduce variance in the Record Prediction step.
>
> ## W3: Model Variation
>
> We intentionally selected O3 as the backbone LLM for the competing methods to ensure they were evaluated under strong and favorable conditions. Providing the AnswerComparison score of competing methods with GPT4o as LLM:
>
> FullCorpus: Hotels- 0.47 (with O3) → 0.24 (GPT4o), WorldCup- 0.44 → 0.49, FinanceBench_agg: 0.04 → 0.04
> VectorPipeline: Hotels: 0.33 → 0.21 , WorldCup: 0.67 → 0.44, FinanceBench_agg: 0.60→ 0.44
>
> > since the HOTELS dataset is synthetic and generated with LLMs, data leakage or stylistic biases may favor schema-driven methods.
>
> This is an interesting point, and we will add it to our limitations. It could be mitigated in multiple ways, for example using multiple generators, and this could strengthen future versions of our benchmark. Nevertheless, because the documents were generated independently, it is unlikely that the performance on aggregative questions can be attributed to data leakage. Specifically, the introduced datasets were generated with GPT-4o, and we show that competing methods do not yield additional gains when using this model.
>
> ## W4: LLM as Judge
>
> To evaluate the correctness of AnswerComparison JLM, we sampled 112 ⟨question, gold answer, predicted answer⟩ triplets generated by different pipelines across multiple datasets. A human annotator then provided a binary judgment for each example, indicating whether the predicted answer conveyed the same information as the gold answer. In 90% of the cases, the human annotator agreed with the JLM’s assessment. We did not repeat this experiment for AnswerRecall JLM, as its scores closely mirror those of AnswerComparison. We will revise Section 5.5 accordingly and provide additional examples in the appendix.
>
> ## W5: Latent Schema Assumption
>
> Indeed the method assumes a single latent schema, which is a strict constraint. However, as exemplified on FinanceBench, this constraint sometimes holds in real-world use-cases (GlobalQA from https://arxiv.org/pdf/2510.26205 is another dataset that holds this assumption, and was published after the paper submission).
>
> We can envision extending the approach to multi-entity corpora by clustering documents during ingestion and applying the ingestion separately to each cluster. At inference time, given a query and a set of schemas, a classifier can determine which schema is most relevant to the query. More complex entities could be supported by allowing ingestion of multiple tables, and we consider exploring those directions as part of our future work.

---

### Official Review · Reviewer_ATaU · 2025-11-05

**Soundness:** 1
**Presentation:** 3
**Contribution:** 1
**Rating:** 2
**Confidence:** 4

**Summary:**

This paper argues that previous RAG studies mainly focus on a much simpler scenario where the answer of the query exists in a small subset of the documents which can be retrieved. This paper studies *aggregative queries* which require retrieving a large set of evidence documents and then reasoning over them. The proposed method is quiet straightforward: prompting the LLM to generate the possible attributes, building the database on the extracted attribute-value pairs, converting the query into SQL for evidence document retrieval and prompting the LLM again for answering. The author also created two synthetic datasets for testing the proposed method.

**Strengths:**

1. This paper focuses on a practical question, aggregative queries, which are frequently encountered in daily life. General RAG methods struggle with this type of query. This study presents a timely approach to address this limitation.
2. The paper's presentation is clear and easy to follow.
3. The datasets are publicly available, which ensure reproducibility.

**Weaknesses:**

1. The *Hotels* dataset is created by prompting the LLM with hand-crafted attributes, so it is well structured to some degree and thus too easy for LLM to guess the attributes. This is different from what the author claims that this work is for unstructured corpus.
2. The synthetic question-answer pairs haven't been verified by human, raising the concerns about the data quality.
3. The proposed method is too straightforward, and there are existing studies with similar methods:

Zhang, Wen, et al. "Trustuqa: A trustful framework for unified structured data question answering." Proceedings of the AAAI Conference on Artificial Intelligence. Vol. 39. No. 24. 2025.

Pinto, David, et al. "Quasm: a system for question answering using semi-structured data." Proceedings of the 2nd ACM/IEEE-CS joint conference on Digital libraries. 2002.

**Questions:**

1. Have you manually verified the generated question-answer pairs of the two synthetic datasets?

---

> ### Author Response · Authors · 2025-11-21
>
> We thank reviewer ATaU for the time and effort invested in reviewing our paper and for raising some questions and concerns. We also appreciate your recognition of the paper’s practical relevance and of the value of the introduced datasets. We now address the raised weaknesses in detail:
>
> ## W1: Hotels Dataset Creation  Method
> The HOTELS corpus was constructed by prompting an LLM to generate a document for each entity, but this does not imply that the resulting corpus is structured; each document consists of unstructured natural language text, as illustrated in Appendix C. The performance of VectorRAG and FullCorpus on this dataset highlights the limitations of existing methods in handling aggregative queries, and the imperfect results of S-RAG further indicate that there remains substantial room for improvement. Although the HOTELS dataset has certain shortcomings, we have demonstrated the advantages of S-RAG on additional synthetic and real-world aggregative queries datasets. Consequently, concerns about a single dataset do not reflect the overall performance or generalizability of the method.
>
> ## W2: Introduced Datasets Quality
> In Section 4.1, we outline our dataset construction methodology: all <query, answer> pairs were first generated programmatically and only then verbalized into natural-language questions. Therefore, accuracy is ensured as long as the natural-language question aligns with the original programmatic query. To further assess correctness, we randomly sampled examples and manually verified their validity. This analysis showed that 84% of the <query, answer> pairs are correct, a level of noise that is comparable to many existing benchmarks.
>
> ## W3: Existing Similar Methods
> The reviewer noted that prior studies propose methods that appear similar; however, these approaches address a fundamentally different problem setting. For example, “TrustUQA: A Trustful Framework for Unified Structured Data Question Answering”
>  introduces a method designed for question answering over multiple types of **structured** data, and provides evaluation on table-based, knowledge-graph, and temporal knowledge-graph QA benchmarks. In contrast, our work focuses on answering aggregative questions over **unstructured** corpus, where the primary challenges—and much of our methodological contribution—lie in the ingestion process and in enabling aggregation over natural text.
> Regarding the 2002 paper “QuASM: A System for Question Answering Using Semi-Structured Data”: it suggests hand crafted heuristics for leveraging document structure—such as tables and section organization—to improve document segmentation (chunking) during ingestion. These methods yield good results for factual question answering. However, they do not address the challenges inherent to aggregative questions, which require reasoning over information distributed across a large number of documents, as demonstrated by the two datasets we introduced.

---

### Official Review · Reviewer_NFjp · 2025-11-08

**Soundness:** 2
**Presentation:** 2
**Contribution:** 2
**Rating:** 4
**Confidence:** 3

**Summary:**

This paper identifies a significant limitation in current RAG (Retrieval-Augmented Generation) systems when handling queries that require aggregating information from a large number of documents. It proposes S-RAG, a method that constructs structured representations during the data ingestion stage and transforms natural language queries into formal queries (e.g., SQL) during inference. The authors also contribute two datasets for aggregative queries—Hotels and World Cup—and demonstrate through experiments that S-RAG significantly outperforms traditional RAG systems and long-context LLMs.

**Strengths:**

Clear problem definition: The paper clearly identifies the limitations of current RAG systems in handling “aggregative queries,” an important real-world scenario.

Strong methodological innovation: S-RAG transforms unstructured documents into a structured database and leverages formal queries for reasoning—a novel and practical approach.

Valuable dataset contribution: The two proposed datasets fill an existing gap in RAG datasets regarding aggregative queries, providing valuable research resources.

Rigorous experimental design: Extensive comparisons are conducted across multiple datasets, including real-world systems (e.g., OpenAI Responses), yielding convincing results.

High practical relevance: The method is well-suited for enterprise private knowledge base scenarios, demonstrating strong potential for real-world deployment.

**Weaknesses:**

Strong methodological assumptions: S-RAG assumes that all documents share the same structure (i.e., a single entity type), which may not hold in real-world multi-entity document scenarios.

Limited generalization: The method’s performance on complex structures (e.g., nested attributes or list-type data) remains unverified, restricting its applicability to more complex corpora.

Small dataset scale: The Hotels and World Cup datasets are relatively small, and the method’s scalability in large-scale industrial settings remains to be validated.

**Questions:**

Can S-RAG be extended to corpora containing multiple entity types (e.g., documents containing both hotel and flight information)?

Could the authors further analyze the types of errors occurring during the record prediction phase and their impact on final answers?

Have the authors considered deeper comparisons with existing structured RAG methods based on knowledge graphs or table extraction?

---

> ### Author Response · Authors · 2025-11-20
>
> We thank reviewer NFjp for the time and effort invested in reviewing our paper and for raising valuable questions and concerns, all of which we took very seriously. We also appreciate the reviewer’s recognition of S-RAG as both innovative and practical for real-world use cases, as well as their acknowledgment of the contribution made by the two new datasets. We address the concerns below:
>
> ## W1: Single Schema Assumption
> We acknowledge, as the reviewer notes, that assuming a single schema can represent all documents in the corpus is indeed a strong requirement. Nevertheless, this assumption does hold in certain real-world scenarios, as demonstrated with the public dataset FinanceBench. GlobalQA (https://arxiv.org/pdf/2510.26205), which was published after our submission, is another dataset that satisfies this requirement.  Furthermore, while we are unable to share these materials publicly or use them directly for research, we are collaborating with multiple leading companies whose internal corpora adhere to the single-schema assumption—for example, regulatory document collections, technical report corpora, and insurance-related corpora.
>
> We can envision relaxing this assumption and extending the framework to corpora containing multiple entities in the following way: During ingestion, the documents could first be clustered, after which the S-RAG ingestion process would be applied independently to each cluster. At inference time, given a query and several candidate schemas (a schema for each cluster), a classifier could determine which schema is most relevant, after which the remainder of the pipeline would proceed as in the original method. More complex entities could also be handled by enabling ingestion into several tables. We see these extensions as promising directions for future work.
>
>
> ## W2 - Limited Generalization
> We agree that our current work does not evaluate generalization to more complex document structures. However, this reflects the scope of the experimental design rather than of S-RAG itself. Even within these settings, we were able to clearly expose the shortcomings of existing methods, such as VectorRag and FullCorpus.
> Extending S-RAG to support more complex document structures is an important direction for future work. We envision addressing this by, for example, splitting the ingestion of a single document into multiple records across tables, or by de-nesting complex objects during the schema prediction phase.
>
> ## W3: Small Dataset Scale
> As for the small size of the introduced datasets, we believe that the low performance of existing methods on these benchmarks already highlights their limitations in handling aggregative questions. We further expect that enlarging the datasets would amplify the spotted challenges competing methods are facing—for example, the FullCorpus method would receive an even smaller proportion of relevant documents as the corpus grows, likely leading to further performance degradation. On the other hand, S-RAG is designed to scale to extremely large datasets: the Schema Prediction step requires only a small sample of documents, and the Records Prediction step operates independently on each document.
>
>
> ## Q2: Records Prediction Error Analysis:
> We conducted an error analysis of the record-prediction results on the HOTELS dataset ingested with the GoldSchema. We manually annotated 70 randomly selected attributes across 10 different records. Only 2% of the attributes were assigned incorrect or values, indicating that this step is highly stable. We repeated the evaluation on the same corpus ingested using the InferredSchema. One source of error in the records prediction process arises from missing attributes (and their associated values), which occur when the inferred schema is incomplete. Manual inspection shows that 33% of the gold-schema attributes were missing in the predicted one. For the attributes included in the inferred schema, the quality of the ingested values was comparable to that achieved when using the GoldSchema. This analysis suggests that the performance degradation of S-RAG under the inferred schema is primarily due to imperfect attribute recall, rather than errors in the Records Prediction step.
>
> ## Q3: Other Structured Methods
> We acknowledge that there are other structured approaches to question answering (see Section 7.1). However, these methods are not well aligned with answering aggregative queries over unstructured corpus. For example, GraphRAG is designed for “sense-making” queries, and other approaches (e.g., https://arxiv.org/pdf/2304.09433) assume much simpler document layout that do not adequately reflect real-world data. Likewise, table-extraction methods are not applicable here, as our documents consist of unstructured text rather than tabular content.

---

### Author Response · Authors · 2025-11-21

We acknowledge, as several reviewers note, that assuming a single schema can represent all documents in the corpus is indeed a strong requirement (it was also discussed in Section 8). Nevertheless, this assumption does hold in certain real-world scenarios, as demonstrated with the public dataset FinanceBench. GlobalQA (https://arxiv.org/pdf/2510.26205), which was published after our submission, is another dataset that satisfies this requirement. From our experience, in practice many companies have internal corpora that satisfy the single-schema assumption, such as regulatory document collections, technical report corpora, and insurance-related corpora.

We can envision relaxing this assumption and extending the framework to corpora containing multiple entities in the following way: During ingestion, the documents could first be clustered, after which the S-RAG ingestion process would be applied independently to each cluster. At inference time, given a query and several candidate schemas (a schema for each cluster), a classifier could determine which schema is most relevant, after which the remainder of the pipeline would proceed as in the original method. We consider exploring these extensions as part of our future work.

---

### Author Response · Authors · 2025-11-21

We thank the reviewers for carefully reading our paper and for their valuable comments, which have helped improve the manuscript. We appreciate their recognition of the importance of studying how QA systems handle aggregative questions, as well as their view of S-RAG as an innovative and practical solution to the challenge such questions pose. We are also grateful for their acknowledgement of the extensiveness of our experimental evaluation, which demonstrates the advantages of S-RAG over the baselines, and for their view of the two newly introduced datasets as valuable resources for future research.
We have submitted a revised version of our paper in which we address the next weaknesses raised by the reviewers:

## Use of different LLMs across systems (Section 6):
We conducted additional experiments in which the existing baselines (FullCorpus and VectorRAG) were experimented with GPT-4o, matching the LLM used by S-RAG. The results show a degradation in their performance relative to the previously reported results obtained with O3.

## Meta-analysis of the automatic metric (Section 5.5, Appendix E):
We evaluated the correctness of the AnswerComparison LLM based metric. In 90% of the randomly sampled cases, the binary judgement of the metric is aligned with the expected outcome.

---

### Meta-Review · Area_Chair_pG53 · 2026-01-05

**Summary:**

This paper introduces **S-RAG**, a framework designed for aggregative queries, where questions require reasoning over a large number of documents. It moves from unstructured text to a structured schema at ingestion and uses text-to-SQL at inference time. The authors contribute two new, but small, datasets (Hotels and World Cup) specifically for this task. Reviewers broadly agree that the paper tackles a practically important, underexplored setting, with a conceptually fresh framing of aggregative QA as structured reasoning. The S-RAG design is clearly presented and empirically outperforms standard RAG and long-context baselines on their new datasets. Key concerns center on the strong single-schema assumption and schema-inference fragility, heavy reliance on LLM prompting, and the limitations of small, synthetic, LLM-generated datasets.

**Reviewer Concerns:**

During the rebuttal stage, the authors successfully provided additional experiments using consistent LLMs across baselines. They validated the "LLM-as-a-judge" metric with human agreement statistics. They also clarified that S-RAG scales efficiently for large-scale ingestion compared to long-context methods. However, several weaknesses persist:
- **Restrictive Assumptions**: The "single-entity/single-schema" assumption is highly restrictive for general-purpose RAG. While the authors proposed a clustering-based extension in the rebuttal, it remains theoretical and unverified (Reviewers NFjp, G5VK, b2z7).
- **Limited Evaluation Scale**: The introduced datasets are small (e.g., 22 documents for the World Cup), which may not surface the indexing and retrieval challenges inherent in massive enterprise corpora (Reviewers b2z7, NFjp).
- **Methodological Incrementalism**: The technical core relies heavily on prompt engineering for SQL generation and extraction, which lacks the innovation in the underlying technical components (Reviewers ATaU, G5VK).

**Reviewer Scores:**

- **Reviewer NFjp (4), ATaU (2), and G5VK (4)**: Likely to lean toward a weak reject or maintain their current scores, as no official acknowledgment of the authors' rebuttals was posted before the cut-off date.
- **Reviewer b2z7 (6)**: Likely to maintain their score.

So the average score is 4.0, which is clearly below the acceptance threshold.

---

### Decision · Program_Chairs · 2026-01-26

Reject